# The Biogenesis of miRNAs and Their Role in the Development of Amyotrophic Lateral Sclerosis

**DOI:** 10.3390/cells11030572

**Published:** 2022-02-07

**Authors:** Jinmeng Liu, Fenghua Zhou, Yingjun Guan, Fandi Meng, Zhenhan Zhao, Qi Su, Weiwei Bao, Xuemei Wang, Jiantao Zhao, Zijun Huo, Lingyun Zhang, Shuanhu Zhou, Yanchun Chen, Xin Wang

**Affiliations:** 1Laboratory of Biochemistry and Molecular Biology, School of Basic Medical Sciences, Weifang Medical University, Weifang 261053, China; 18353687185@163.com; 2Neurologic Disorders and Regenerative Repair Laboratory, Weifang Medical University, Weifang 261053, China; zhoufh@wfmc.edu.cn (F.Z.); guanyj@wfmc.edu.cn (Y.G.); zly199311@126.com (L.Z.); 3Department of Pathology, School of Basic Medical Sciences, Weifang Medical University, Weifang 261053, China; 4Department of Histology and Embryology, School of Basic Medical Sciences, Weifang Medical University, Weifang 261053, China; 18800460605@163.com (F.M.); jinqiuhupan@163.com (Z.Z.); suqi5976@163.com (Q.S.); jssybaoweiwei96@163.com (W.B.); 18863662168@163.com (X.W.); zjt15233694339@163.com (J.Z.); zijunhuo1998@163.com (Z.H.); 5Department of Orthopedic Surgery, Brigham and Women’s Hospital, Harvard Medical School, Boston, MA 02115, USA; szhou@bwh.harvard.edu; 6Department of Neurosurgery, Brigham and Women’s Hospital, Harvard Medical School, Boston, MA 02115, USA

**Keywords:** amyotrophic lateral sclerosis, miRNAs, noncoding RNA, biomarker, therapy, neurodegeneration

## Abstract

Amyotrophic lateral sclerosis (ALS) is a neurodegenerative disease that affects upper and lower motor neurons. As there is no effective treatment for ALS, it is particularly important to screen key gene therapy targets. The identifications of microRNAs (miRNAs) have completely changed the traditional view of gene regulation. miRNAs are small noncoding single-stranded RNA molecules involved in the regulation of post-transcriptional gene expression. Recent advances also indicate that miRNAs are biomarkers in many diseases, including neurodegenerative diseases. In this review, we summarize recent advances regarding the mechanisms underlying the role of miRNAs in ALS pathogenesis and its application to gene therapy for ALS. The potential of miRNAs to target diverse pathways opens a new avenue for ALS therapy.

## 1. Introduction

Amyotrophic lateral sclerosis (ALS) is a chronic disease caused by selective invasion and progressive degeneration of motor neurons (MNs). Its main clinical manifestations are progressive muscle bundle fibrillation, muscle weakness and atrophy [1]. It ultimately results in death in 3–5 years, with a prevalence of 2–3/100,000 [2,3]. Currently, there is still a lack of therapeutic drugs to cure this disease [4,5]. Riluzole and edaravone are currently approved treatments [6]. Riluzole prolongs patient survival for 2–3 months at most with little effect on the quality of life, while edaravone slightly improves the activity of patients, but the impact on survival remains unclear [7,8].

ALS includes sporadic ALS (sALS) and familial ALS (fALS). Approximately 5~10% of cases of ALS are familial cases. Since 1990, more than 25 mutated genes related to ALS have been found [9]. The mutant genes include SOD1, FUS, C9ORF72, MATR3, TARDBP, PFN1, OPTN, TUBA4A and TBK1 [10,11]. The dominant genetic mutation of SOD1 gene was first described and was associated with 15% of fALS cases [12]. As these genes indicate the key disease pathways for treatment, it is now more possible to develop better treatments for ALS patients [13,14]. The pathogenesis of ALS is complex and involves many factors [15]. Although pathogenic genes have been screened, the mechanism of MN degeneration is still not completely clear. At present, it is considered related to a long list of factors, among which are disturbances in RNA metabolism [16], protein misfolding and aggregation [17], marked neuromuscular junction (NMJ) abnormalities [18], immune system deficiency [19], nucleocytoplasmic transport defects [20], impaired DNA repair [21], excitotoxicity [22], mitochondrial dysfunction [23], oxidative stress [24], cytoskeletal derangements [25], axonal transport disruption [26], neuroinflammation [27], oligodendrocyte dysfunction [28] and vesicular transport defects [29].

MicroRNAs (miRNAs) are endogenous small RNAs of about 18–25 nucleotides; they are single-stranded, short and highly conserved and do not encode proteins. miRNAs play mainly a regulatory role in vivo and inhibit gene expression at the post-transcriptional level by base pairing with the 3- or 5-terminal untranslated region of mRNA. It is estimated that about one-third of human gene expression may be regulated by miRNAs [30,31]. One miRNA can interact with several or multiple genes. Hundreds of miRNAs are known to play important roles in brain development and pathology [32,33]. Because most miRNA expression is highly tissue- and cell-specific, miRNAs can be used in disease diagnosis and treatment. miRNAs can regulate more than 50% of coding genes, making them central to the stability of biological processes. Therefore, it is speculated that miRNAs might participate in neurodegenerative diseases [34,35,36].

Many studies have shown that miRNAs participate in nervous system development [37,38]. For example, miRNAs are essential for axonal regeneration of spinal MNs in the ablation of Dicer in mouse sciatic nerve cells [39]. Otaegi and colleagues found that miRNAs are involved in the development of the spinal cord and the differentiation of neuronal subtypes [40]. Meanwhile, miRNAs participate in regulating gene expression of the physiological processes of many cells, including cell death [41,42]. Moreover, Zhou and colleagues found that miRNA-124 is associated with ALS [43]. In conclusion, miRNAs play irreplaceable roles in various processes of spinal MNs [44]. Numerous studies have also proven that miRNAs have a unique role in ALS pathogenesis [45].

In this review, we provide a brief description of miRNA biogenesis and the potential involvement of miRNAs in the development of ALS. We summarize interventions in the expression of a variety of miRNAs that may be relevant to the pathophysiology of ALS. We also discuss miRNAs as the targets of therapeutic intervention by regulating and/or correcting their abnormal activity in ALS for clinical treatment.

## 2. miRNA Biosynthesis and Function

An increasing number of studies have found that various types of noncoding RNA (ncRNA) can be used as biomarkers, including tRNA, rRNA, piwi-RNA and miRNA. The latest version of the miRNA database (miRBase) has cataloged 434 miRNAs in *Caenorhabditis elegans*, 466 in *Drosophila melanogaster* and 2588 in humans [46].

Most mature miRNA genes are located in functional gene coding regions or noncoding regions and may be expressed in clusters or independently. The biogenesis of miRNAs is conducted by two RNase III enzymes, Drosha and Dicer, which catalyze two subsequent processing events in the nucleus and cytoplasm [47]. In the nucleus, genomic DNA is transcribed to generate a long primary miRNA (pri-miRNA), which is then cleaved by Drosha to form a precursor miRNA (pre-miRNA) with hairpin structure, about 70 nucleotides (nt) long. These hairpin RNAs are transported to the cytoplasm by nuclear export receptor exportin-5 and are cleaved by Dicer in the cytoplasm to form mature miRNAs of 19–23 nt. In addition, functional miRNAs can also be generated through noncanonical miRNA biogenesis pathways. These include mirtrons that are generated via pre-mRNA splicing and miRNAs generated from small nucleolar RNA (snoRNA) precursors [46]. Mature single-stranded miRNAs bind to a series of proteins to form miRNA-induced silencing complexes (miRISCs), which in turn bind to the 3′-UTR region of the target mRNA, preventing the translation of the bound mRNA or directly degrading the target miRNA. Each miRNA can regulate multiple target genes, and specific target miRNAs can be simultaneously regulated by multiple miRNAs (Figure 1) [48,49,50,51].

On the basis of fully understanding their biogenesis and function, miRNAs have become the main focus of many studies, including studies on cancer, cardiovascular disease, neurodegenerative diseases and retinal disorder [52].

## 3. miRNAs and ALS

### 3.1. Circulating miRNA Biomarkers in ALS

ALS is a multisystem disease that is difficult to diagnose. Thus far, many researchers have mostly focused on exploring the role of miRNAs in the accurate diagnosis, prognosis or monitoring of the progression of ALS [53,54]. The miRNA signatures in biological samples from ALS patients, such as circulating body fluid, cell components or muscle biopsy, are detected by high-throughput methods [55,56,57,58,59].

#### 3.1.1. Serum miRNAs in ALS

Support for pathological molecular intervention before the onset of ALS is largely lacking. The examination of sALS and fALS serum revealed the gene expression profile of miRNAs in ALS. Freischmidt’s team was the first to provide miRNA signatures of both fALS patients and asymptomatic ALS mutation carriers. They reported 24 miRNAs downregulated in premanifest ALS, 91.7% of which overlapped with fALS patients. Meanwhile, the most significant GDCGG and SGGC motifs were independently identified as highly enriched in downregulated miRNAs in fALS patients and ALS mutation carriers [60]. Nevertheless, Freischmidt and colleagues found miR-1825 and miR-1234-3p were consistently downregulated in sALS patient serum [61]. This study found changes in homogeneous miRNA in different types of ALS and the consistent sequence motif of downregulated miRNA. It suggested that different ALS genes can induce the same pathophysiological pathway. It also provided a basis for exploring new therapeutic targets and offers a strong foundation for evaluating the treatment of presymptomatic disease-modification of ALS.

Another study compared the levels of serum miRNAs in 20 sALS and 3 fALS patients with patients of Alzheimer’s disease (AD) and multiple sclerosis and healthy controls. The details are as follows: seven miRNAs (miR-1, miR-19a-3p, miR-133b, miR-133a-3p, miR-144-5p, miR-192-3p and miR-192-5p) were upregulated and six miRNAs (miR-139-5p, miR-320a, miR-320b, miR-320c, miR-425-5p and let-7d-3p) were downregulated in ALS patients compared to both healthy and ill controls [62]. This study correlated changes in miRNA expression with clinical parameters over time, which will transform these miRNAs into diagnostic and prognostic markers and provide new insights into disease development.

#### 3.1.2. Muscle-Specific miRNAs in ALS

Muscle-specific miRNAs (myo-miRNAs) may participate in regulating the ALS disease process, where upper and lower MN degeneration leads to muscle atrophy. To prove the hypothesis, Pegoraro et al. measured myo-miRNAs levels by real-time PCR in diagnostic muscle biopsies in 18 ALS patients: eight genetic forms (C9orf72-ALS and SOD1-ALS), five sALS cases and five ALS cases affected only by upper MN disease. Analysis of myo-miRNAs found miR-1 downregulated in SOD1-ALS and sALS patients, miR-206 upregulated in the skeletal muscle of C9orf72-ALS and SOD1-ALS patients, miR-133a downregulated in sALS and UMN patients and miR-133b downregulated in sALS patients compared to healthy controls [63]. This study enables us to better understand different types of miRNA expression levels in ALS and their role in neurodegenerative diseases.

Moreover, Russell et al. measured and compared the miRNA expression levels in skeletal muscle. miR-23a, miR-29b, miR-206 and miR-455 were significantly increased by 91%, 59%, 89% and 91% (respectively) in ALS patients compared to healthy controls. miR-31 was increased in ALS patients and neurogenic disease (ND) by 101% and 79%, respectively. Notably, only miR-23a was significantly higher in the ALS than the ND patients [64]. These results suggest that miR-23a directly targets and reduces skeletal muscle PGC-1α protein levels and, along with findings that miR-23a damages mouse skeletal muscle mitochondrial function, identify a new therapeutic target for ALS.

#### 3.1.3. Leukocyte miRNAs in ALS

In order to detect changes in specific miRNAs related to ALS in the most readily available human tissues, some studies have examined the changes of miRNA expression profile in leukocytes.

De Felice’s team detected the expression levels of eight miRNAs in ALS patient samples. miR-149, miR-328-5p, miR-451, miR-583, miR-638, miR-665 and miR-1275 were downregulated across genders and in all specimen types tested. Interestingly, miR-338-3p was upregulated in all the sALS samples compared to unaffected controls [65]. This is consistent with a previous observation regarding the miR-338-3p expression in the brain of ALS patients [66]. These results provide further evidence that expression changes of specific miRNAs in peripheral blood are a distinctive feature of sALS. In addition, these miRNAs may help to elucidate the pathogenic mechanisms of sALS and have the potential to become biomarkers.

Vrabec et al. determined differential expression of 10 miRNAs in blood sample leukocytes from 84 sALS patients. Nine of the ten miRNAs were significantly upregulated across the patient cohort: let-7b, miR-9, miR-124a, miR-132, miR-206, miR-338, miR-451, miR-638 and miR-663a [67]. These findings pave the way for miRNAs to be used as diagnostic biomarkers for ALS.

### 3.2. The Mechanism of miRNAs in ALS

A lack of accurate targets has always hindered the treatment of ALS, and identifying signaling pathways and cellular mediators remains an enormous challenge in finding new therapeutics. However, each changed miRNA represents a potential new gene biomarker for clinical diagnosis (Figure 2) [68,69,70,71,72,73,74].

The discovery of these miRNAs provides a basis for in-depth study of the pathogenesis, accurate diagnosis and effective treatment of ALS. To address the lack of targets, scientists have conducted a series of mechanism studies over the years, as shown in Figure 3.

#### 3.2.1. miRNA and NMJ Abnormalities in ALS

Williams’ team initially explored the skeletal muscle miRNA expression during ALS progression in symptomatic SOD1^G93A^ transgenic mice, finding that miR-206 is dramatically induced. They then bred miR-206^−/−^ mice expressing a low copy number of SOD1-G93A. First, they found that loss of miR-206 did not affect the onset of disease but did accelerate progression and shorten survival by one month. They went on to find that miR-206 offsets the deleterious effects of histone deacetylase 4 (HDAC4) on reinnervation following injury. After further verification, these results suggest that miR-206 promotes and HDAC4 prevents NMJ innervation via opposing effects on fibroblast growth factor binding protein 1 (FGFBP1). In summary, their results indicate that miR-206 is a regulatory factor of ALS pathogenesis. The beneficial effects of miR-206 are mediated by muscle-derived factors, which promote nerve–muscle interaction during MN injury [75]. Since prior studies have identified miR-206 as a reliable biomarker [76,77], miR-206 is expected to be a new therapeutic target for ALS based on physiologic reinnervation self-healing responses.

#### 3.2.2. miRNA and Neuroinflammation in ALS

Another widely studied miRNA is the inflammatory miR-155, which is significantly increased in fALS and sALS patients and the presymptomatic SOD1^G93A^ mice [78,79,80]. Butovsky and colleagues also identified the key role of miR-155 in SOD1^G93A^ mice. Genetic ablation of miR-155 reduced the expression of APOE in the spinal cord of SOD1^G93A^ mice. This result demonstrates that the APOE pathway is a main proinflammatory mediator of microglia during disease progression in SOD1^G93A^ mice. Therefore, it may also be a potential target for the treatment of human ALS [80].

Parisi’s team found that the miR-125b–A20–NF-κB axis is associated with maintaining inflammatory signals in microglia. Specifically, inhibition of miR-125b reduces the transcriptional targets expression of NF-κB and protects MNs from death induced by G93A microglia activation. In light of this study, the miR-125b–A20 axis may be activated in ALS, thus playing a crucial role in inflammatory pathology mediated by NF-κB signal, effectively balancing “good” and “bad” neuroinflammation in ALS [81].

#### 3.2.3. miRNA and Neuronal Survival and Apoptosis in ALS

Knowing that miR-193b-3p is downregulated in ALS patients, Li and colleagues found that miR-193b-3p expression level was also decreased in SOD1^G93A^ mice. To investigate the role of miR-193b-3p, they transfected hybrid mouse motor neuron-like cells (NSC-34 cells) with miR-193b-3p mimics or inhibitors. The results demonstrated that upregulation of miR-193b-3p promotes cell death in NSC-34 cells. Further studies verified that miR-193b-3p directly targeted tuberous sclerosis 1 (TSC1) and regulated mechanistic target of rapamycin complex 1 (mTORC1) activity in NSC-34 cells. Meanwhile, miR-193b-3p negatively regulates autophagy in NSC-34 cells. In conclusion, this research revealed that reducing miR-193b-3p is required for cell survival by improving autophagy through the TSC1–mTOR pathway. It might provide a key target of early therapeutic strategies in ALS [82].

Rohm et al. assessed whether N-myc downstream-regulated gene 2 (NDRG2) and miR-375-3p are prospective targets for studying sALS pathogenesis in the wobbler mouse model. They found miR-375-3p and NDRG2 deregulation in the spinal cord during the disease. Both miR-375-3p and NDRG2 are associated with p53 in the relationship summarized as the following cascade: The rise in levels of NDRG2 leads to increased formation of ROS, which is a stress signal activating p53. Due to the downregulation of miR-375-3p, the inhibition of p53 is ineffective, resulting in overexpression of NDRG2, increasing the generation of ROS and creating a vicious circle. Therefore, increased miR-375-3p might be a protective mechanism to prevent cell apoptosis by decreasing p53 expression and maintaining the NDRG2 level [83].

Kim and colleagues first discovered that miR-18b-5p expression level decreased in mtNSC-34 cells, which was related to altered Hif1α, miR-18b-5p, miR-206, mctp1, Rarb and Mef2c. Their findings can be generalized as follows: (a) decreased miR-18b-5p regulated Hif1α expression post-transcriptionally, (b) Hif1α increased Mef2c as a transcription factor, (c) Mef2c greatly increased miR-206 expression, (d) miR-206 degraded both mctp1 and Rarb, (e) decreased mctp1 inhibited Ca^2+^ signal, (f) decreased Rarb prevented neuronal cell differentiation and (g) decreased miR-18b-5p promoted apoptotic cell death in ALS SOD1 mutation. These results provide a new understanding of the downregulation of miR-18b-5p-dependent pathogenesis of ALS. More importantly, this new mechanism will likely have great clinical significance for the treatment of ALS, after further research [84].

Li et al. found that miR-183-5p was involved in the coupling of stress sensing and scavenging. Their cellular experiments demonstrated that miR-183-5p regulates apoptotic and necroptotic pathways by directly targeting RIPK3 and PDCD4, thus identifying miR-183-5p as a regulator of programmed neuron death. Overexpression of miR-183-5p increased neuron survival in stress situations, while its knockout led to neuron death. Their study supports the current viewpoint that cell stress and cell death/survival are linked by common mechanisms, both regulated by miRNA, which provides a new target for future ALS interventions [85].

miR-124 is the most abundant miRNA in the central nervous system (CNS) and an important regulator of neurogenesis, which is central to synaptic plasticity, the neurite growth process and neuron–astrocyte differentiation. miR-124 is also thought to be a negative regulator of inflammation. Its upregulation in MNs of SOD1^G93A^ mice is related to neurodegeneration and microglial activation. To explore miR-124 function in MN degeneration, Vaz and colleagues used NSC-34 cells overexpressing human wild-type (WT) SOD1 or G93A mutated SOD1 (m-SOD1). Transfection of pre-miR-124 into WT cells resulted in decreased MN activity and increased cell death through early apoptosis, similar to the discovery of mSOD1 MNs. In contrast, treating mSOD1 MNs with anti-miR-124 completely eliminated cell death. Moreover, normal levels of miR-124 can prevent the damage of neurite growth, axon transport, synaptic signal, mitochondrial dynamics, microglia polarization into proinflammatory phenotype by the secretome and pathogenicity of spinal organotypic in miR-124-enriched MNs. This study provided a new perspective for targeted therapy of ALS, especially in its early stages [86].

#### 3.2.4. miRNA and Cytoskeletal Derangements in ALS

The aggregation of intermediate filaments in MNs is a symbol of ALS pathogenesis. Hawley found that downregulated miR-9 and miR-105 in the spinal cord of ALS patients targeted the 3′-UTRs of three intermediate filaments, INA, NEFL and PRPH, to regulate gene expression. Overall, their results suggest that miR-9 and miR-105 are necessary to maintain the stoichiometry of intermediate filaments. Therefore, the loss of miRNAs might lead to the imbalance of intermediate filaments and affect the normal function of ALS MNs. Restoration of the expression of these miRNAs in ALS patients would be expected to reduce changes in intermediate fiber stoichiometry to slow down the progress of the disease [87].

Axon degeneration and NMJ interruption are key events in ALS pathology. NMJ is the first damaged compartment in ALS. This disease is considered to be a distal axonopathy in the non-cell-autonomous process [88]. Recent studies have shown that non-neuronal cells have been recognized as critical players in MN death and survival [89]. Maimon and colleagues found decreased miR-126-5p expression in presymptomatic male ALS SOD1^G93A^ mice, and miR-126-5p targets axon destabilizing type 3 semaphorins (Sema3A). Overexpression of miR-126-5p temporarily rescued axonal degeneration and NMJ damage in vitro and in vivo. Therefore, they demonstrated that this change in miR-126-5p promoted the non-cell-autonomous mechanism of ALS MNs degeneration, suggesting it as a novel mechanism of ALS pathogenesis [90].

In earlier studies by Helferich et al., miR-1825 was the only downregulated miRNA in serum of sALS and fALS patients, and a decreasing trend of miR-1825 was also observed in premanifest ALS mutation carriers. In recent experiments, they proved that the decrease in miR-1825 resulted in the upregulation of translation of tubulin-folding cofactor b (TBCB). In addition, they discovered that excessive TBCB results in depolymerization and degradation of tubulin alpha-4A (TUBA4A), which is encoded by a known ALS gene. Overall, the miR-1825–TBCB–TUBA4A regulatory axis was dysregulated in both sALS and fALS patients. Further research into disturbances of the microtubule cytoskeleton associated with this regulatory axis will contribute important information to our understanding of ALS pathogenesis [91].

Table 1 summarizes the potential circulating miRNA biomarkers of ALS.

### 3.3. miRNA Treatment in ALS

#### 3.3.1. miRNA Therapeutic Approaches

With the development of gene expression detection technology, many different types of miRNAs have been identified. Studies have revealed the regulatory functions of some molecules in vitro and in vivo, and their regulatory targets can be predicted by computational tools and bioinformatics algorithms. The studies showed that abnormal expression of miRNA is related to disease progression. Some preclinical studies have been conducted to develop miRNA-based therapies [92]. In recent years, many researchers have assessed the expression level and possible role of miRNAs in ALS animal models and the potential of miRNAs for ALS treatment [56,74,93]. Currently, there are two main strategies for miRNA therapy, namely antisense technology and miRNA replacement therapy. When an upregulated miRNA contributes to the pathology of disease, its expression can be blocked using antisense technology. The anti-miRNA oligonucleotides (AMOs) bind to target miRNAs through complementary bases to inhibit miRNAs in antisense technology. Another treatment strategy is miRNA replacement therapy. These miRNA mimics contain exactly the same sequence as mature endogenous miRNAs, enabling them to complex with RISC and target mRNA to supplement deficient miRNA levels in disease [94].

Currently, numerous approaches for ALS treatment have been developed, including ASOs, CRISPR/Cas9 and base editing [95,96,97]. Delivery methods based on miRNA therapy mainly include viral-based and non-viral-based systems. Viral-based systems have the advantage of high infection efficiency and high continuous expression of miRNAs or antagomirs. Compared with viral systems, nonviral systems have lower toxicity and immunogenicity and no restriction on the transferred DNA size, through transfection efficiency is low [98]. Viral-based systems include retrovirus, lentivirus, adenovirus and adeno-associated virus. Nonviral systems include lipid- and polymer-based systems and inorganic carriers [99]. Recent studies have shown the safety and feasibility of miRNAs in ALS treatment, making miRNA therapy a promising path forward [100].

#### 3.3.2. miRNA Therapy for ALS

To test the inhibition of miRNA in the CNS as a new treatment for ALS, Koval and colleagues developed oligonucleotide-based miRNA inhibitors (anti-miRs) that can inhibit miRNA in the entire CNS and periphery by intracerebroventricular injection. In this research, anti-miR-155 treatment of SOD1^G93A^ mice prolonged the survival for 10 days and the course of disease for 15 days, while treatment of control anti-miR did not affect the survival or disease course. Thus, they demonstrated for the first time that antisense oligonucleotides may successfully inhibit miRNA in the whole brain and spinal cord. Therefore, miR-155 may be a new therapeutic target for ALS. This also remains a promising direction for the treatment of ALS [79].

Nolan’s team explored the function of miR-29a in mediating the progression of ALS disease. They found that expression of miR-29a was highly upregulated in the spinal cord of SOD1^G93A^ mice during disease progression and confirmed anti-interference mediated miR-29a knockdown in the brain and spinal cord of SOD1^G93A^ mice. Although they found no significant changes in motor function and lifespan after miR-29a gene knockout in the progression or pathology of ALS disease, they demonstrated an increased lifespan trend in male SOD1^G93A^ mice after induction of miR-29a knockdown at postnatal day 70. The results suggest that miR-29a might be a marker of disease progression in SOD1^G93A^ mice and provide the basis for the therapeutic regulation of miR-29a in ALS [101].

Many studies have provided strong evidence that silencing the expression of mutant SOD1 protein can prolong the survival of SOD1-linked ALS mice. Borel et al. reported a silencing method based on artificial miRNA named miR-SOD1, which is transmitted using the adeno-associated virus rAAVrh10 system. rAAVrh10 is a serotype with proven safety in the CNS from clinical trials [102]. In adult SOD1^G93A^ transgenic mice, silencing SOD1 significantly delayed the onset and death of SOD1^G93A^ mice and preserved muscle strength as well as motor and respiratory function. This study also found that intrathecal delivery of the same rAAVrh10-miR-SOD1 significantly and safely silences SOD1 in lower MNs in nonhuman primates. All these findings support that rAAVrh10-miR-SOD1 deserves further development to treat human SOD1-linked ALS [103].

AAV vectors are very efficient for gene transfer into the CNS, where they mediate long-term gene expression without apparent toxicity. The latest-generation AAV vector based on AAV9 and AAVrh10 capsid can efficiently transduce various types of cells in the spinal cord, including MNs. Because misfolded hSOD1 is widely expressed in neurons, astrocytes, microglia and oligodendrocytes, this extensive tropism in the spinal cord is particularly important. Stoica and colleagues studied the therapeutic effect of the AAV9 vector encoding hSOD1-specific amiR (amiR^SOD1^) injected into the lateral ventricle of neonatal SOD1^G93A^ mice. AAV9 transmitted amiRSOD1 to CNS cells and reduced the expression of hSOD1. Treatment with AAV9-amiR^SOD1^ can improve the survival of neonatal SOD1^G93A^ mice, delay the onset of paralysis, improve the integrity of axons and the number of MNs and delay the occurrence of spinal cord inflammation. Overall, Stoica’s team demonstrated gene silencing as a treatment for SOD1^G93A^ mice. They achieved a 50% increase in median survival and significant preservation of motor function by using AAV9-amiR^SOD1^ vector to deliver intracerebral ventricular (ICV) neonatally. This study demonstrates that the delivery of artificial miRNA targeting hSOD1 by using AAV9 provides the greatest survival benefit of any AAV gene therapy in SOD1^G93A^ mice or rats [104].

Gene therapy mediated by recombination adeno-associated virus (rAAV) is a promising therapeutic approach for ALS. Previous studies have confirmed the therapeutic potential of mutant SOD1-RNAi transmitted by intrathecal (IT) injection of rAAV. However, delivery needs to be improved to overcome the differences in transduction efficiency and therapeutic effect. Li and colleagues studied the effect of injection speed on CNS transduction efficiency. They found that slow injection enhanced conduction in the spinal cord relative to the brain and surrounding organs, while fast injection results in higher conduction in the brain and surrounding organs relative to the spinal cord. To examine how these findings affect the results of RNAi treatment, they injected rAAVrh10-GFP-amiR-SOD1, an rAAV vector expressing GFP and an artificial miRNA targeting SOD1, by slow and fast injection. Both injection speeds were effective, but the slow injection was better than the fast injection. These results suggest that the speed of injection affects the gene transmission advantage of different CNS sites, which should be considered in future treatment trials involving intrathecal injection [105].

Previous studies have shown that miR-17~92 might be valuable as a prognostic marker of MN degeneration and a candidate therapeutic target for SOD1-ALS [106,107,108]. According to the results using the double reporter embryonic stem cell (ESC) differentiation system developed in their study, Tung and colleagues found that miR-17~92 was downregulated in SOD1^G93A^ spinal lateral motor column MNs (LMC-MNs) before MN loss, accompanied by the accumulation of nuclear phosphatase and tensin homolog deleted on chromosome 10 (PTEN). To determine the potential role of miR-17~92 for ALS treatment, they overexpressed miR-17~92 in SOD1^G93A^ mice MNs through a gene method and virus-mediated gene therapy. Both methods significantly improved exercise defects and prolonged lifespan. Overexpression of miR-17~92 rescued human SOD1^+/l144f^ MNs, and IT of adeno-AAV9-miR-17~92 improved the motor defects and survival of SOD1^G93A^ mice. In view of the emerging application of miRNAs as biopharmaceuticals, they assumed that the discovery of survival-promoting miRNAs would provide valuable insights for the treatment and development of ALS. More importantly, the salvage effect of miR-17~92 further elucidates miRNA application in precision medicine of neurodegenerative diseases [109].

Martier and colleagues previously revealed the latent capacity of artificial anti-C9orf72-targeting miRNA (miC) to decrease the toxicity caused by repeat-containing transcripts. They tested the AAV5-miC efficiency of silence in iPSC neurons and ALS mice. They proved the feasibility of silencing C9orf72 repeat-containing transcripts by gene therapy based on miRNAs. First, they found that AAV5 effectively transduces miRNA into neuronal and non-neuronal cells. Secondly, AAV5-miC reduces the number of repeat-containing C9orf72 transcripts in iPSC neurons and effectively silences C9orf72 in the nucleus of iPSC neurons. Last but not least, both aav5-miC32 and AAV5-miC46 reduced C9orf72 and RNA foci in Tg (C9orf72_3) line 112 mice. Therefore, they indicated that AAV5-miC transduces different types of neuronal cells and reduces the accumulation of repeat-containing C9orf72 transcripts. In addition, they confirmed that the silence of C9orf72 in the nucleus and cytoplasm has additional value for the treatment of ALS patients. Overall, the recent study demonstrates that AAV-miC targets C9orf72 in the nucleus and is expected to relieve the toxicity of RNA-mediated ALS [110].

The SOD1^G93A^ transgenic ALS mouse model can gradually develop progressive dyspnea and dysphagia in the late stage of disease. Lind and colleagues inferred that directly treating the tongue of SOD1 mice could improve respiratory and swallowing functions and prolong survival. AAVrh10 miR-SOD1 was injected into SOD1^G93A^ mice at 6 weeks of age to target sublingual MNs. These neurons provide the motor function to all muscles of the tongue and play a critical role in breathing and swallowing. The results showed that miR^SOD1^ prolonged the survival of female mice, but not male mice. Meanwhile, IT injection of miR^SOD1^ improved respiratory function but did not improve the swallowing function of MT mice. This study suggests that IT AAVrh10 miR-SOD1 treatment might help to maintain respiratory function by increasing upper respiratory tract patency rather than swallowing. As a promising treatment, it is worthy of further exploration [111].

Mueller and colleagues performed intrathecal infusion of AAV-miR-SOD1 gene therapy in two patients with SOD1-mediated ALS. Patient 1 developed meningoradiculitis after infusion. The SOD1 level in spinal cord tissue of patient 1 at autopsy was lower than that in untreated patients with SOD1-mediated ALS and healthy controls. In addition, the strength of the right leg improved briefly and was relatively stable for the course of the disease. However, those researchers could not be sure that SOD1 inhibition played a role in this patient’s clinical process, as the improvement of function may have reflected recovery from meningoradiculitis. Patient 2 was pretreated with immunosuppressive drugs and showed no meningoradiculitis. Although patient 2 received no clinical benefit from viral vector therapy, the ALS composite score was stable and the patient’s lung activity was stable within 12 months. His functional status and life ability were relatively stable for 60 weeks. Because this process may typically slow disease progression in SOD1-ALS patients, it is impossible to draw clinical conclusions on the treatment effect. This study suggested that IT infusion of AAV-delivered miRNA can be accomplished. The therapeutic effect of viral vector-mediated gene suppression on ALS patients with SOD1 mutations requires more research [112].

The studies of miRNAs in the treatment of ALS are summarized in Table 2.

## 4. Conclusions and Perspectives

In the vast majority of ALS patients, the etiology of ALS is unclear. Although the pathogeny of some fALS patients is related to a wide range of gene mutations, some related genes are still not fully elucidated. Most genotypes may not predict phenotypes [5]. Although many pathological mechanisms of ALS have been studied, it is still a fatal disease with no effective treatment. The heterogeneity of the disease and the lack of satisfactory treatment confirmed that ALS is a multifactorial and multisystem disease [113]. Since the discovery of the SOD1 mutation in 1993, the genetic components of ALS have been deeply explored. This continuing exploration will reveal the mechanism(s) leading to motor neuron death, contribute to clinical diagnosis and classification and provide personalized guidance for treatment [5]. Therefore, it is necessary to better understand the pathogenic mechanism and investigate the underlying pathological relationship between various cellular processes in order to ensure effective treatment.

As endogenous genes of the human genome, miRNAs have vital roles in the development and cell survival, proliferation and differentiation. Regulatory mechanisms of miRNAs are crucial in cell maintenance, homeostasis and response to dynamic environments [114,115]. Abnormal miRNA expression and function caused by genomic alterations or miRNA biogenesis changes, such as mutations, deletions, amplifications, transcriptional changes and enzymatic differences, are key factors in disease development and progression [46].

As an increasingly powerful tool, gene therapy may provide treatment for the pathogenesis in various neurodegenerative diseases. In the aging population, it is necessary to improve treatment methods; gene therapy may fill this gap. Based on this, Foust et al. conducted seminal and key research in the field of ALS gene therapy based on AAV. In their late-onset model, a single peripheral injection of AAV9 encoding shRNA significantly slowed disease progression and prolonged survival after therapeutic administration [116]. In recent years, the progress of vector design and delivery technology has greatly improved our ability to target the brain and spinal cord, paving the way for successful gene therapy experiments in preclinical models and clinical trials. In addition, in order to better illuminate the basic mechanisms of underlying neuronal death, ongoing research will also identify new targets for gene therapy [56,117].

As previously shown, miRNAs have been utilized to make significant progress in the clinical treatment of ALS. However, there are still some challenges in the clinical development of miRNA delivery, including the inefficient penetration or off-target effect of miRNA antagonists or mimics on tissues, the possible induction of immunotoxicity and neurotoxicity and the dysfunction of therapeutic miRNAs. To date, a variety of nanocarriers with different structures, sizes, shapes, chemical properties and functions have been developed for the intracellular delivery of miRNAs. These delivery systems are based on one or more molecular components and are specifically designed to overcome major delivery barriers, including biocompatibility, endosomal escape, effective miRNA binding/protection, cell-specific targeting for increased bioavailability or unreliable stability in circulation [92,94].

Although miRNA detection and intervention techniques are maturing, many problems still need to be solved before ALS can be effectively treated: (1) Many evolutionarily conserved miRNAs come from different gene families, and their functional connections need to be further explored. (2) miRNAs may have different target genes, or a gene might be the recognized target of various miRNAs. How do we select the target genes that play a role? (3) For different mutant genotypes and sporadic cases, can the same miRNA achieve the same therapeutic effect? (4) How should we select the appropriate miRNA delivery materials based on the variety of carrier materials recently developed? (5) What are the correct criteria for selecting single-dose or multidose injection for treatment and the dose itself? (6) How can the treatment effect be effectively evaluated?

The availability of miRNA in the treatment of ALS will be improved through solving this series of problems and questions. The selection of gene therapy targets, vectors and delivery technologies must be based on successful gene therapy experiments and translated into clinical practice. We can expect the future to bring miRNA-based products used as diagnostic and therapeutic tools.

## Figures and Tables

**Figure 1 cells-11-00572-f001:**
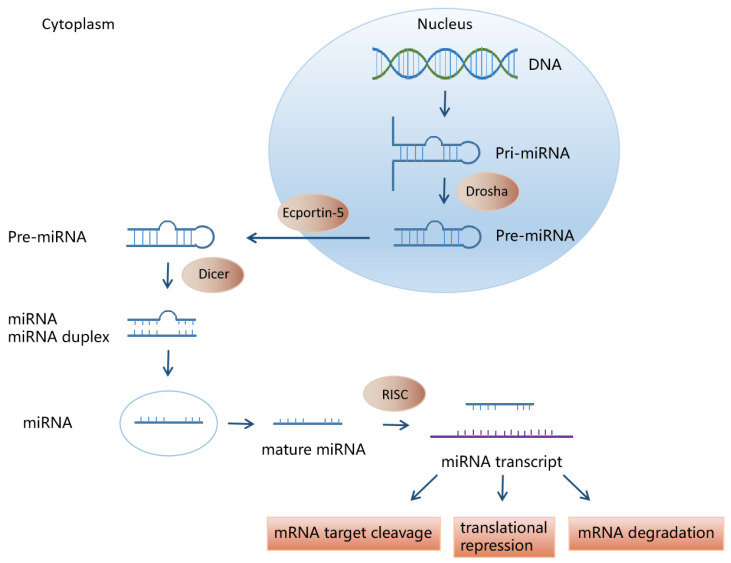
miRNA biosynthesis process. miRNA is processed by RNA polymerase Ⅱ to pri-miRNA. Then, pri-miRNAs are cleaved into pre-miRNAs by complex microprocessors including Drosha. Afterward, pre-miRNAs are exported from the nucleus to the cytoplasm through exportin-5, where they are further dissected by Dicer into double-stranded miRNA. After binding to RISC, the miRNA double strand is unchained by the Argonaute protein. The mature strand remains in RISC and binds to target mRNA for gene regulation, while the other passenger strand is hydrolyzed.

**Figure 2 cells-11-00572-f002:**
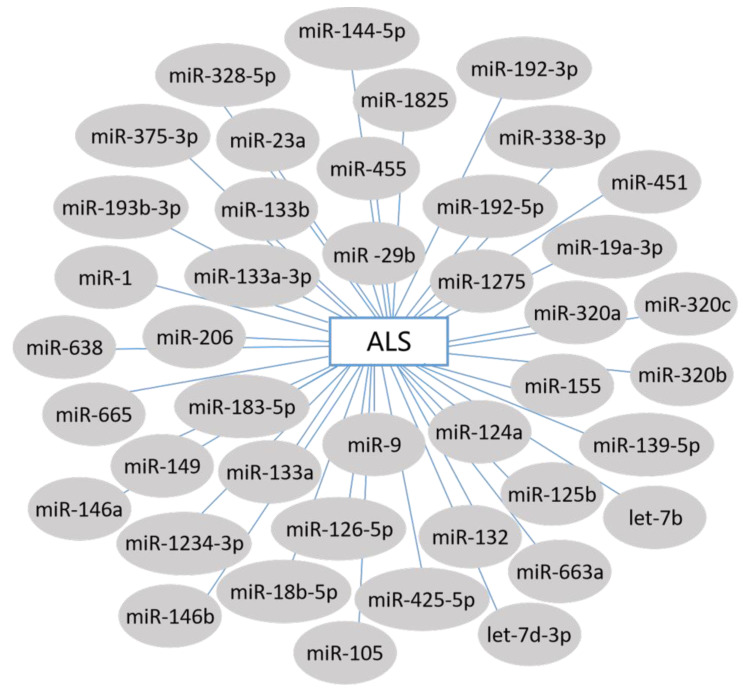
ALS-related miRNAs. The miRNAs associated with ALS are presented in gray boxes. Blue lines indicate that the miRNAs are involved in ALS.

**Figure 3 cells-11-00572-f003:**
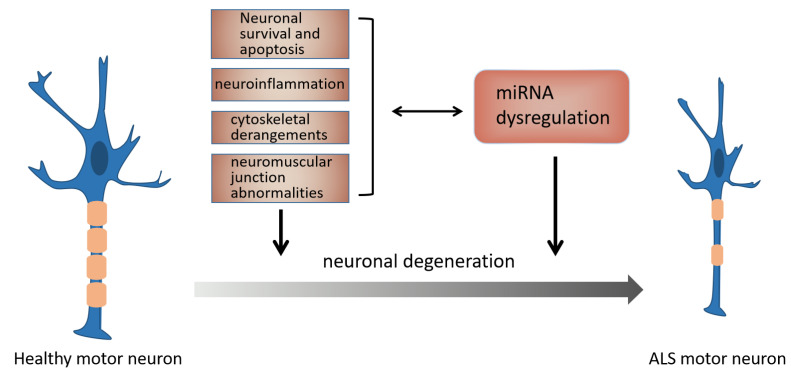
The dysregulation of miRNA function and pathophysiological mechanisms in ALS may be reciprocally caused and synergistically contribute to neuronal degeneration.

**Table 1 cells-11-00572-t001:** Potential circulating miRNA biomarkers in ALS.

miRNA	Model	Change	Target/Signaling Pathway	Functions
miR-206 [75]	SOD1^G93A^ mice	↑	HDAC4 FGFBP1	miR-206 slowed progression of ALS by sensing MN damage and promoting compensatory regeneration of neuromuscular synapses.
miR-155 [80]	SOD1^G93A^ mice	↑	APOE pathway	Genetic ablation of miR-155 reversed the proinflammatory signature of both peripheral tissues.
miR-125b [81]	SOD1^G93A^ mice	↑	NF-κB pathway A20	miR-125b inhibition through A20 protein protects MNs from death induced by activating G93A microglia.
miR-193b-3p [82]	SOD1^G93A^ mice NSC-34 cell	↓	mTOR TSC1	Downregulation of miR-193b-3p is essential for cell survival by targeting TSC1–mTOR signaling in NSC-34 cells.
miR-375-3p [83]	wobbler mouse	P0: ↑ P20:↑ P40:↓	p53 NDRG2	After downregulating miR-375-3p expression, inefficient inhibition of p53 results in overexpression of NDRG2, increasing ROS generation and creating a vicious cycle.
miR-18b-5p [84]	fALS patient SOD1^G93A^ mice NSC-34 cell		Hif1α Mef2c miR-206 Mctp1 Rarb	miR-18b-5p induced HIF1α, which increased the expression of Mef2c. Mef2c upregulated miR-206 as a transcription factor. Inhibition of mctp1 and RARB as miR-206 targets induced intracellular Ca^2+^ levels and reduced cell differentiation, respectively.
miR-183-5p [85]	SOD1^G93A^ mice		PDCD4 RIPK3	miR-183-5p regulated cell apoptosis by targeting PDCD4 and necroptosis by RIPK3, respectively.
miR-124 [86]	SOD1^G93A^ mice NSC-34 cell			Upregulation of miR-124 is related to the degeneration of mSOD1 MNs, the deregulation of neuro-immune crosstalk and the imbalance of homeostasis.
miR-105 miR-9 [87]	sALS patient	↓	NEFL PRPH, INA	Downregulation of miR-9 and miR-105 in sALS might contribute to the loss of intermediate filament stoichiometry, ultimately leading to intermediate filament aggregation and eventually neuronal death.
miR-126-5p [90]	SOD1^G93A^ mice	↓	Sema3A	Downregulation of miR-126-5p levels facilitated axon degeneration and NMJ disruption.
miR-1825 [91]	sALS patient	↓	TBCB CASP3	Downregulation of miR-1825 caused translational upregulation of TBCB, which might lead to depolymerization and degradation of TUBA4A.

**Table 2 cells-11-00572-t002:** miRNA-based therapeutics in ALS.

miRNA	Model	Approach Type	Therapeutic Efficacy
miR-155 [79]	SOD1^G93A^ mice	anti-miR-155	Inhibition of miR-155 extended survival for 10 days and disease duration for 15 days (38%).
miR-29a [101]	SOD1^G93A^ mice	miR-29a-specific antagomir	miR-29a knockout did not show significant changes in disease progression or pathology of ALS, but there was a trend of prolonged lifespan and delayed disease onset in male mice.
miR-SOD1 [103]	SOD1^G93A^ mice	rAAVrh10-miR-SOD1	Silencing SOD1 significantly delayed the onset and death of SOD1^G93A^ mice and preserved muscle strength as well as motor and respiratory function.
miR-SOD1 [104]	SOD1^G93A^ mice	AAV9-amiR^SOD1^ vector	Treatment of AAV9-amiR^SOD1^ vector prolonged median survival by 50% and delayed hindlimb paralysis.
miR-SOD1 [105]	SOD1^G93A^ mice	rAAVrh10-GFP-amiR-SOD1	Slow injection of rAAVrh10-GFP-amiR-SOD1 to silence SOD1 has better therapeutic effect than rapid injection.
miR-17~92 [109]	SOD1^G93A^ mice	AAV9- miR-17~92	Overexpression of miR-17~92 in adult MNs can delay MN degeneration, enhance motor function and prolong lifespan in SOD1-linked ALS.
miR-C9orf72 [110]	Tg(C9orf72_3) Line 112 mice	AAV5-miC	After treatment, repeat-containing c9orf72 transcripts and RNA foci were significantly reduced.
miR-SOD1 [111]	SOD1^G93A^ mice	AAVrh10-miR-SOD1	Intralingual miR-SOD1 injection extends survival and improves respiratory function in mice.
miR-SOD1 [112]	SOD1 fALS patient	AAV-miR-SOD1	Two patients with SOD1-mediated ALS were injected with AAV-miR-SOD1, which achieved short-term therapeutic effect. The results showed that intrathecal miRNA could be a potential treatment for SOD1-mediated ALS.

## Data Availability

Not applicable.

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
