# Peer review of "The Biogenesis of miRNAs and Their Role in the Development of Amyotrophic Lateral Sclerosis"

_cells, 2022, doi:10.3390/cells11030572_

Round 1
Reviewer 1 Report
The present manuscript reviews the role of micro-RNAs in ALS pathophysiology and its potential therapy. The review is comprehensive and informative. The manuscript is well organized and structured. The authors open with findings correlating various miRNAs to ALS in patients and animal models and discuss their potential use as biomarkers. The authors then describe the possible mechanistic role of the various miRNA in the pathophysiology of ALS. Finally, the authors describe the potential use of miRNA for treating ALS, either by antogomirs delivery or by miRNA mimicking. The authors wisely use diagrams and tables to summarize each section in the manuscript.
Bellow, please find some minor comments and suggestion:
- In section 3.2.1 Pegoraro et al. is described as measuring miRNA expression in western blot…. However, it was HDAC protein that was measured by western blot while miRNA levels were measured by real-time PCR.
- Line 313-315 “Therefore, they demonstrated 313 that this change in miR-126-5p promoted the non-cellular autonomous mechanism of 314 MN degeneration in ALS…” The sentence is not clear… what does the phrase “non-cellular autonomous mechanism” means?
- In section 3.3.2 in describing Koval et al. experiments it would be beneficial to mention that targeting miRNA to the CNS was achieved by intra-cerebroventricular infusion.
- The opening statement in line 389 about the efficiency and safety of AAV vectors in the CNS requires a reference.
- The opening statement in line 421 about previous studies demonstrating the potential of miR-17-92 as a prognostic marker in ALS requires citing these studies at the end of the sentence.
Author Response
Dear Editor,
We appreciated the editor to give us the opportunity to revise the manuscript as a minor revision. We thank very much for the reviewers’ constructive suggestions which would significantly improve the quality of the manuscript. With the added information and modified table, we hope that the revised manuscript meets the approval of the reviewers, and is of the quality and impact to merit publication in Cells. Here, we submit a revised manuscript, which we believe has been significantly improved.
The new changes to respond reviewers’ comments have been marked with blue color. The following is a point-to-point response to the reviewers’ comments. In addition, regarding the duplicated parts of ithenticate report, we have modified related sentences/texts by marking them with gray background.
Reviewer #1:
Major comments:
Question 1. In section 3.2.1 Pegoraro et al. is described as measuring miRNA expression in western blot…. However, it was HDAC protein that was measured by western blot while miRNA levels were measured by real-time PCR.
Answer: Thanks for reviewer 1’s suggestion. we have modified the experimental method from western blot to real-time PCR in section 3.1.2 in the revised manuscript (line 161).
Question 2. Line 313-315 “Therefore, they demonstrated 313 that this change in miR-126-5p promoted the non-cellular autonomous mechanism of 314 MN degeneration in ALS…” The sentence is not clear… what does the phrase “non-cellular autonomous mechanism” means?
Answer: ALS is considered a complex disease, with unique MN features as well as non-cell-autonomous contributions. Some evidence suggests that the NMJ is the first compartment to be disrupted in ALS rather than the MN soma; the disease is recognized as distal axonopathy in a non-cell-autonomous process. We have added a description of non-cell-autonomous mechanisms on lines 311-312.
Question 3. In section 3.3.2 in describing Koval et al. experiments it would be beneficial to mention that targeting miRNA to the CNS was achieved by intra-cerebroventricular infusion.
Answer: We have added the contents of injection methods (lines 365-366).
Question 4. The opening statement in line 389 about the efficiency and safety of AAV vectors in the CNS requires a reference.
Answer: We have added a reference (95) in the revised manuscript (line 388).
Question 5. The opening statement in line 421 about previous studies demonstrating the potential of miR-17-92 as a prognostic marker in ALS requires citing these studies at the end of the sentence.
Answer: We have added references (99-101) in the revised manuscript (line 428).
Reviewer 2 Report
Liu et al summarize the role of miRNAs in ALS and strategies involving miRNAs (either as targets or as effectors for regulating target gene expression) to treat ALS. The review is well written. My major concern is a lack of references for various topics and the need to better highlight the wide range of therapeutics that have been, and are, under development for ALS.
- Lines 58-64, the authors should cite primary references for each example of dysregulation provided for ALS.
- Lines 65-70, there is not sufficient references supporting the primary literature findings for miRNA.
- Line 122, the following sentence needs more context “On the basis of fully understanding its biogenesis and function, miRNA has become 122 the main focus of most studies.” What are the “most studies” the authors are referring to?
- In the interest of fairness, numerous approaches for SOD1-ALS have been developed, including ASOs, which have progressed through clinical trials (Miller et al. N Engl J Med 2020; 383:109-119 DOI: 10.1056/NEJMoa2003715) and CRISPR-Cas9 (Gaj et al. Sci Adv. 2018 DOI: 10.1126/sciadv.aar3952) and base editing (Lim et al. Mol. Ther. 2020 Apr 8;28(4):1177-1189. doi: 10.1016/j.ymthe.2020.01.005.). These alternate approaches should be highlighted at some point in the text.
- Foust et al. Mol. Their. 2013 Dec;21(12):2148-59. doi: 10.1038/mt.2013.211 should be discussed and cited. It is seminal and a key study in the field of AAV-based gene therapies for ALS, including those involving miRNAs.
Author Response
Dear Editor,
We appreciated the editor to give us the opportunity to revise the manuscript as a minor revision. We thank very much for the reviewers’ constructive suggestions which would significantly improve the quality of the manuscript. With the added information and modified table, we hope that the revised manuscript meets the approval of the reviewers, and is of the quality and impact to merit publication in Cells. Here, we submit a revised manuscript, which we believe has been significantly improved.
The new changes to respond reviewers’ comments have been marked with blue color. The following is a point-to-point response to the reviewers’ comments. In addition, regarding the duplicated parts of ithenticate report, we have modified related sentences/texts by marking them with gray background.
Reviewer #2:
Major comments:
Question 1.Lines 58-64, the authors should cite primary references for each example of dysregulation provided for ALS.
Answer: Thanks for reviewer 2’s suggestion. We have cited primary references (14-28) accordingly.
Question 2. Lines 65-70, there is not sufficient references supporting the primary literature findings for miRNA.
Answer: We have added references (29-30) supporting the primary literature findings for miRNA.
Question 3.Line 122, the following sentence needs more context “On the basis of fully understanding its biogenesis and function, miRNA has become 122 the main focus of most studies.” What are the “most studies” the authors are referring to?
Answer: The “most studies” include research on cancer, cardiovascular disease, neurodegenerative diseases, retinal disorder, and so on. We have added the contents about “most studies” in lines 124-125.
Question 4. In the interest of fairness, numerous approaches for SOD1-ALS have been developed, including ASOs, which have progressed through clinical trials (Miller et al. N Engl J Med 2020; 383:109-119 DOI: 10.1056/NEJMoa2003715) and CRISPR-Cas9 (Gaj et al. Sci Adv. 2018 DOI: 10.1126/sciadv.aar3952) and base editing (Lim et al. Mol. Ther. 2020 Apr 8;28(4):1177-1189. doi: 10.1016/j.ymthe.2020.01.005.). These alternate approaches should be highlighted at some point in the text.
Answer: We have added these important approaches in section 3.3.1 of the revised manuscript (lines 351-353).
Question 5. Foust et al. Mol. Their. 2013 Dec;21(12):2148-59. doi: 10.1038/mt.2013.211 should be discussed and cited. It is seminal and a key study in the field of AAV-based gene therapies for ALS, including those involving miRNAs.
Answer: We have discussed and cited Foust et al’ article in section 4 (lines 517-520).